# Can Large Language Models Think Like Doctors? An Interactive Approach to Evaluating Clinical Reasoning

## Abstract

Clinical diagnosis begins with doctor-patient interaction, during which physicians iteratively gather targeted information , determine examination and refine their differential diagnosis through patients' response. This interactive clinical-reasoning process is poorly represented by existing LLM benchmarks that focus on question-answering or multiple-choice format. In this work, we propose **iClinReason**, an interactive framework for assessing clinical reasoning in LLMs through simulated diagnostic dialogues. Grounded in a disease knowledge graph, our method dynamically generates patient cases with structured symptom profiles, and instantiates a patient agent that engages in a multi-turn diagnostic conversation with the target LLM, which acts as a doctor agent. Our evaluation protocol goes beyond diagnostic accuracy by incorporating fine-grained efficiency analysis and a rubric-based assessment of diagnostic quality across multiple dimensions. Experimental results reveal that iClinReason effectively exposes critical clinical reasoning gaps in state-of-the-art LLMs, offering a more nuanced and clinically meaningful evaluation paradigm.

## 1 Introduction

Large language models (LLMs) have demonstrated increasing potential in healthcare applications, including clinical decision support, patient-facing chatbots, and automated medical documentation (Omiye et al., 2024; McDuff et al., 2025; Falcetta et al., 2023). As these systems move closer to integration within real-world clinical environments, the need for rigorous and clinically meaningful evaluation of their reasoning capabilities becomes critical(Cabral et al., 2024; Goh et al., 2024). Existing benchmarks (Jin et al., 2020; Hendrycks et al., 2021; Zuo et al., 2025) have largely focused on static evaluations, such as multiple-choice exams or single-turn question answering. While useful for measuring factual recall, these formats fail to capture the interactive and iterative nature of clinical reasoning.

In actual clinical practice, diagnosis often begins with medical consultation and involves active information gathering, continual refinement of differential diagnoses, and evidence integration (Brush & Brophy, 2017; Gruppen et al., 1991). This interactive, stepwise process unfolds over time through structured dialogue, where doctors iteratively inquire symptoms, assess risk factors, rule in or out conditions, and justify diagnostic decisions. It requires not only breadth of medical knowledge but also higher-order clinical reasoning skills, including contextual interpretation, logical inference, adaptive hypothesis testing, and effective communication. Consequently, the benchmarks that evaluate models solely on final-answer accuracy risk overestimating their clinical readiness, as they ignore the coherence and quality of the reasoning trajectory. Moreover, the static nature of evaluation datasets often introduces information leakage or benchmark contamination (Xu et al., 2024; Chen et al., 2025), which further undermines the reliability of current evaluation paradigms.

To address these challenges, we propose iClinReason, an interactive framework for assessing clinical reasoning in LLMs through simulated multi-turn doctor-patient diagnostic dialogues. iClinReason moves beyond static test formats by modeling the diagnostic process as a dynamic conversation between three agents: a patient agent that simulates symptom disclosure and response to inquiry, an examiner agent that provides medical examination reports, and a doctor agent instantiated by the

target LLM under evaluation. Notably, our framework is grounded in a structured disease knowledge graph, enabling controlled generation of diverse patient cases with nuanced symptom profiles and plausible differential diagnoses. Through this interactive setup, iClinReason captures key aspects of the clinical reasoning process, including hypothesis generation, test ordering, adaptive revision of differential diagnoses, and diagnostic efficiency. Our evaluation protocol extends beyond result-based metrics by incorporating process-oriented analysis and a fine-grained, rubric-based assessment of diagnostic quality across multiple dimensions, such as logical consistency, differential breadth, and cognitive flexibility, providing a more comprehensive and clinically grounded appraisal of model behavior.

In summary, the main contributions of this work are threefold:

- **Interactive evaluation framework for clinical reasoning:** We propose iClinReason, an interactive diagnostic framework that simulates multi-turn doctor-patient conversations to evaluate LLMs' clinical reasoning.

- **Knowledge-grounded case generation and process-aware assessment:** We develop a disease knowledge graph-driven pipeline for generating diverse and dynamic patient cases, and introduce process-aware metrics that evaluate both diagnostic efficiency and quality.

- **Comprehensive analysis of LLMs' diagnostic behaviors:** We evaluate state-of-the-art LLMs using iClinReason, uncovering systematic reasoning deficiencies, such as inadequate information seeking and limited revision of differential diagnoses, and providing actionable insights for developing more reliable clinical LLMs.

## 2 RELATED WORKS

**LLM in Medical Domain**  In recent years, LLMs have demonstrated significant potential in the medical field, with applications spanning clinical decision support, patient-facing interactions, and automated medical documentation (Singhal et al., 2023; Tu et al., 2025; Wang et al., 2023). These models can assist clinicians by analyzing vast amounts of medical literature and health records to support diagnosis and treatment planning (Maity & Saikia, 2025; Busch et al., 2025; Xie et al., 2024; Zhang et al., 2023). Studies have explored the performance of LLMs on real-world clinical cases, finding that their diagnostic accuracy can be comparable to that of human physicians (Singhal et al., 2025). However, the application and evaluation of LLMs in medicine still face considerable challenges (Artsi et al., 2025). One main concern is that these models can generate incorrect or misleading information. This is especially risky in healthcare (Han et al.). Therefore, it is essential to develop strong evaluation methods to ensure LLMs are safe and reliable before they are used in clinical practice.

**Medical Benchmarks**  The evaluation of LLMs in the medical domain has largely relied on static question answering benchmarks that primarily measure factual recall and domain knowledge coverage. For instance, MedQA (Jin et al., 2020) evaluates models on USMLE-style multiple-choice questions, offering a measure of medical knowledge. PubMedQA (Jin et al., 2019) moves beyond exam questions by formulating research-level QA questions derived from biomedical literature, requiring models to infer answers from textual evidence. While these benchmarks have advanced the measurement of medical knowledge in LLMs, they remain primarily static, single-turn, and exam-style. Most focus on final-answer accuracy or lexical overlap metrics (e.g., BLEU, ROUGE) rather than on the dynamic reasoning trajectories that are central to real clinical decision-making. Recent efforts such as MedCaseReasoning (Wu et al.) and MedR-Bench Qiu et al. (2025) are constructed from case reports and aim to evaluate how models infer diagnoses from structured descriptions. However, these benchmarks remain static in nature and do not capture the interactive dynamics of doctor–patient communication. To reflect clinical practice, several datasets have been created with multi-turn doctor–patient dialogues (Liu et al., 2025; Yu et al., 2024). AIPatient (Yu et al., 2024) is an simulated patient system powered by six specialized LLM agents and a knowledge graph built from real medical data, designed to generate patient interactions for medical training and AI evaluation. However, these studies primarily focus on simulated multi-turn medical dialogues, and lack a fine-grained evaluation of the model's clinical reasoning capabilities. To bridge the gap, we propose iClinReason, an interactive framework that evaluates LLMs through simulated doctor–patient dialogues, tracking not just what they diagnose, but how they reason along the way.

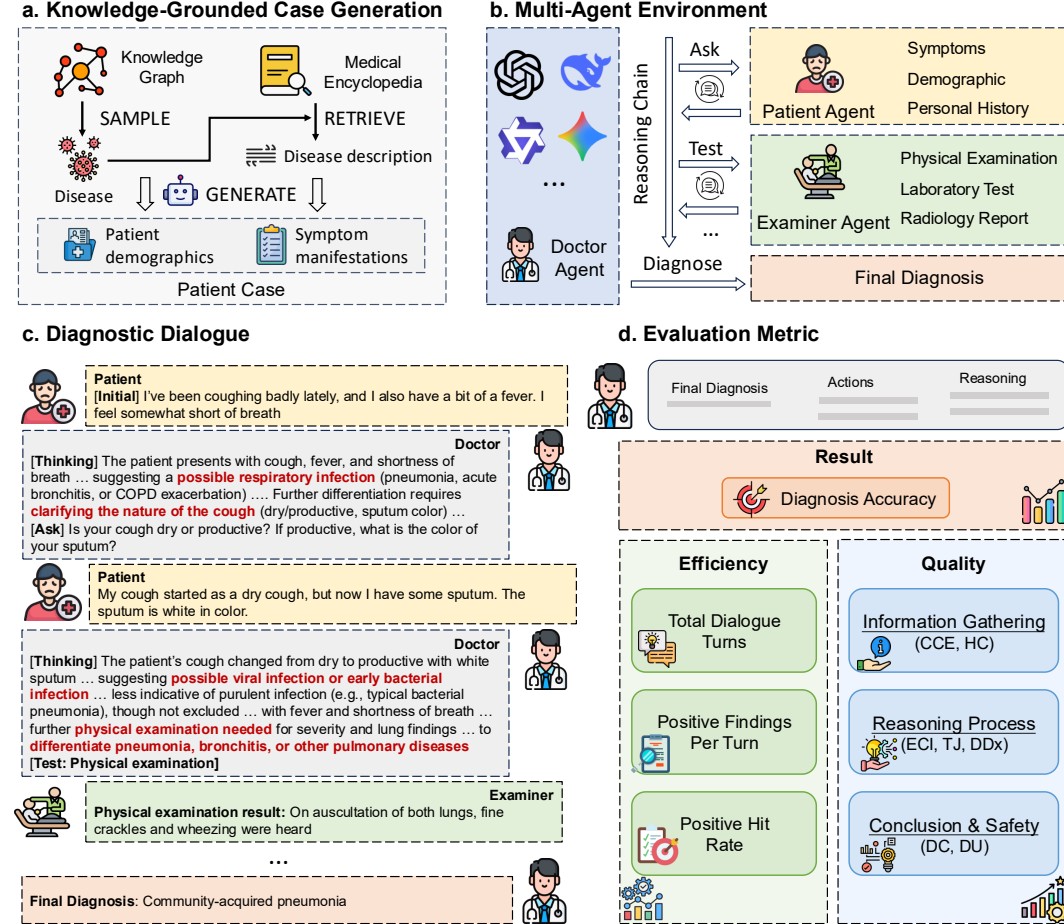

Figure 1: Overview of iClinReason. **(a)** Knowledge-grounded case generation combines a disease–symptom knowledge graph with medical encyclopedia text to synthesize patient cases. **(b)** A multi-agent environment models diagnostic consultation, where the Doctor Agent (LLM being evaluated) interacts with Patient and Examiner Agents through actions Ask, Test, and Diagnose. **(c)** A diagnostic dialogue example illustrates information gathering, hypothesis revision, and evidence integration, leading to a final diagnosis. **(d)** Evaluation metrics encompass diagnostic accuracy, efficiency, and quality.

## 3 METHODS

In this section, we introduce the proposed iClinReason, an interactive evaluation framework with three main components: (1) knowledge-grounded case generation, (2) a multi-agent environment for diagnostic dialogue, and (3) a comprehensive evaluation protocol, as illustrated in Figure 1.

### 3.1 KNOWLEDGE-GROUNDED CASE GENERATION

A central goal of iClinReason is to construct evaluation cases that are both diagnostically faithful and resistant to contamination from pretraining corpora. To this end, we formulate case generation as a mapping

$$\mathcal{G} : (K_G, K_E, \Theta) \mapsto \mathcal{C}, \tag{1}$$

where $K_G$ is a structured disease–symptom knowledge graph, $K_E$ is an unstructured medical encyclopedia, $\Theta$ is a generative LLM, and $\mathcal{C}$ is the resulting case profile. For each diagnostic session, $\mathcal{G}$ first samples a disease node $d \in K_G$ and retrieves the corresponding descriptive passage $T_d \subset K_E$. Conditioned on $(d, T_d)$, the generator $\Theta$ synthesizes patient demographics $\mathcal{P}_{\text{info}}^d$ and symptom manifestations $\mathcal{S}_d$ under a set of medical consistency constraints derived from the topology of $K_G$. This

dynamic and knowledge-grounded sampling ensures that each case is coherent with domain expertise while being previously unseen, thus mitigating the risk of memorization by evaluated models. The output of this process is a structured case profile

$$\mathcal{C} = \big(d,\, T_d,\, P_{\text{info}},\, \mathcal{S}_d\big). \tag{2}$$

This profile serves as the latent ground truth from which the patient agent simulates dialogue, the examiner agent provides test results and against which the doctor agent's reasoning is evaluated.

Compared with static, hand-curated clinical vignettes, the dynamically generated cases offer complementary advantages: (1) *Contamination resistance*, since the space of possible cases is huge and instantiated dynamically at evaluation time; (2) *Knowledge-grounded consistency*, because symptoms and attributes are sampled under explicit structural constraints of $K_G$ and thus remain clinically coherent; and (3) *Controlled clarity*, as all case elements are derived from curated sources rather than noisy records, making the evaluation less confounded by missing or ambiguous information. Notably, we emphasize that real-world clinical cases remain the gold standard; our contribution lies in providing a controlled, contamination-free method for generating cases.

### 3.2 MULTI-AGENT ENVIRONMENT FOR DIAGNOSTIC DIALOGUE

The design of iClinReason is inspired directly by the structure of real-world medical consultations, where physicians, patients, and clinical examination systems interact in complementary ways. To capture this interactive reasoning process, we introduce a multi-agent environment that formalizes diagnostic dialogue as a structured interaction among agents. This paradigm goes beyond conventional static QA settings by simulating the iterative and multi-source evidence gathering that underpins clinical reasoning.

**Agent Roles** Before modeling the dialogue, we first define the key actors in the environment. Real clinical encounters involve multiple participants with clearly delineated responsibilities, and we mirror this division by assigning each role to a dedicated agent

$$\mathcal{A} = \{A_D,\, A_P,\, A_E\}, \tag{3}$$

corresponding to the doctor, the patient, and the examiner. The Doctor Agent $A_D$ represents the LLM under evaluation and serves as the *sole deliberative agent*, tasked with identifying the ground-truth disease $d$ through iterative information gathering and reasoning. In contrast, the Patient Agent $A_P$ and Examiner Agent $A_E$ function as *responsive environmental simulators*: $A_P$ provides symptom reports and demographic information grounded in the case profile $\mathcal{C}$, while $A_E$ delivers laboratory test results and imaging reports upon request. Together, these three agents define the interactive ecosystem within which diagnostic reasoning is tested.

**Dialogue Dynamics** The consultation is modeled as a temporally ordered sequence of utterances, capturing the evolving state of the dialogue:

$$H_t = (u_1, u_2, \ldots, u_t), \quad t = 1, 2, \ldots, T. \tag{4}$$

Here, $H_t$ encodes all information exchanged up to time $t$, serving as the shared memory across agents. The dialogue is initialized by the Patient Agent $A_P$, which generates the first utterance $u_1$ containing the chief complaint based on the symptom manifestation $\mathcal{S}_d \in \mathcal{C}$:

$$u_1 = A_P(\mathcal{S}_d). \tag{5}$$

For each subsequent turn, the active agent generates the next utterance conditioned on the $H_{t-1}$ and the case profile $\mathcal{C}$. This recursive structure ensures that reasoning is path-dependent, reflecting the way physicians revise hypotheses based on accumulating evidence rather than isolated inputs.

**Agent Actions** To emulate structured clinical reasoning, the Doctor Agent $A_D$ operates within a discrete action space:

$$\mathcal{A}_D = \{\texttt{Ask},\, \texttt{Test},\, \texttt{Diag}\}, \tag{6}$$

and at each turn, it produces an action and the associated outcomes based on the dialogue history:

$$(a_t^D, o_t^D) = A_D(H_{t-1}), \quad a_t^D \in \mathcal{A}_D. \tag{7}$$

The action `Ask` produces an outcome $o_t^D = q_t$, which corresponds to a natural language query aimed at eliciting subjective information about symptoms or history. `Test` results in $o_t^D = r_t$, denoting a request for an objective examination. `Diag` outputs $o_t^D = d_t$, representing the model's final diagnostic decision that terminates the interaction. By abstracting physician behavior into these three canonical moves, we isolate the fundamental reasoning primitives that drive clinical consultations while keeping the evaluation tractable and reproducible.

In contrast, the Patient $A_P$ and Examiner $A_E$ agents are deterministic simulators that do not possess an autonomous action space. Their behavior is a reactive response to the Doctor's action $a_t^D$, governed by the case profile $\mathcal{C}$. We define them as part of the environment's response mechanism.

**Environment Response** Once the Doctor Agent issues $(a_t^D, o_t^D)$, the environment determines which simulation agent (i.e., $A_P$ or $A_E$) responds and produces the utterance $u_t$:

$$u_t = \begin{cases} A_P(q_t \mid \mathcal{C}), & \text{if } a_t^D = \texttt{Ask} \wedge o_t^D = q_t, \\ A_E(r_t \mid \mathcal{C}), & \text{if } a_t^D = \texttt{Test} \wedge o_t^D = r_t, \\ \texttt{End}, & \text{if } a_t^D = \texttt{Diag} \wedge o_t^D = d_t. \end{cases} \tag{8}$$

Each response $u_t$ is strictly constrained by the structured case profile $\mathcal{C}$, ensuring consistency between the simulated dialogue and the ground-truth disease $d$. In practice, this means that subjective responses from $A_P$ always align with the symptom set in $\mathcal{C}$, while test results generated by $A_E$ reflect medically plausible outcomes tied to $d$. This design guarantees that any observed errors in reasoning are attributable to the Doctor Agent rather than noise in the environment.

**History Update** After each interaction, the dialogue history is updated recursively:

$$H_t = H_{t-1} \oplus (o_t^D, u_t). \tag{9}$$

This operation appends the doctor's action outcome and the corresponding response to the evolving dialogue state. By feeding $H_t$ back into the next decision of $A_D$, the framework naturally captures the iterative nature of hypothesis refinement. The process continues until either a diagnosis is stated via $a_t^D = \texttt{Diag}$ or the maximum turn limit $T_{\max}$ is reached, ensuring bounded yet realistic interactions. The resulting dialogue trajectory provides a complete trace of the reasoning process, which is later used for both outcome-based and process-oriented evaluation.

### 3.3 EVALUATION METRICS

To comprehensively assess the clinical reasoning capabilities of LLMs, we designed a multi-faceted evaluation protocol. Inspired by Objective Structured Clinical Examinations (OSCE) widely used in medical education (Fu et al., 2025), this protocol moves beyond measuring only the final diagnostic *accuracy* to also scrutinize the *efficiency* and *quality* of the diagnostic process. This enables deeper insights into how LLMs gather information, revise hypotheses, and justify decisions, mirroring the cognitive workflow of expert clinicians. Specifically, the evaluation metrics are described as follows.

**Diagnostic Accuracy** This dimension focuses on the accuracy of the final diagnosis, measuring whether the Doctor Agent correctly identifies the patient's disease. It serves as a fundamental measure of the model's clinical decision-making, forming the basis for further evaluation of diagnostic efficiency and quality.

**Diagnostic Efficiency** This dimension evaluates how effectively the Doctor Agent gathers clinically relevant information and progresses toward the final diagnosis. It is measured along three complementary indicators:

1. **Total Dialogue Turns:** The overall number of turns required to reach a diagnosis, reflecting efficiency in information gathering and decision-making.

2. **Positive Findings Per Turn:** The positive findings per case, indicating how much clinically useful information is elicited in each dialogue.

3. **Positive Hit Rate:** The proportion of positive findings among all findings, capturing the precision of information gathering by minimizing irrelevant or non-contributory data.

---

**Algorithm 1:** Interactive Clinical Reasoning (iClinReason) Evaluation Framework

---

**Input:** Structured medical knowledge graph $K_G$, Unstructured medical encyclopedia $K_E$, Patient information generator $\Theta$, Doctor Agent $A_D$ (LLM to be evaluated), Patient Agent $A_P$, Examiner Agent $A_E$, Maximum dialogue turns $T_{\max}$, Evaluation metrics $\mathcal{M}$
**Output:** Diagnostic accuracy, efficiency, and reasoning quality score

   // Step 1: Dynamic Case Generation
1 Sample a disease node $d \sim K_G$;
2 Retrieve disease-specific descriptive text $T_d \leftarrow \text{Query}(d, K_E)$;
3 Generate patient demographics $P_{\text{info}}$ and symptom set $S_d$ using $\Theta$;
4 Construct a case profile $\mathcal{C} = (d, T_d, P_{\text{info}}, S_d)$;

   // Step 2: Multi-Agent Initialization
5 Initialize $A_P$, $A_E$, and $A_D$;
6 Initialize dialogue history $H_0 = \emptyset$;
7 $u_1 \leftarrow A_P(S_d)$;         // Patient Agent initiates the dialogue
8 $H_1 \leftarrow H_0 \oplus u_1$;
9 $t \leftarrow 1$;

   // Step 3: Diagnostic Dialogue Loop
10 **while** $t < T_{\max}$ ***and** no diagnosis stated* **do**
11    $(a_t^D, o_t^D) \leftarrow A_D(H_{t-1})$;     // Doctor Agent makes an action based on $H_{t-1}$
12    **if** $a_t^D = Ask \wedge o_t^D = q_t$ **then**
13       $u_t \leftarrow A_P(q_t \mid \mathcal{C})$;
14    **else if** $a_t^D = Test \wedge o_t^D = r_t$ **then**
15       $u_t \leftarrow A_E(r_t \mid C)$;
16    **else if** $a_t^D = Diag \wedge o_t^D = d_t$ **then**
17       **break**;
18    $H_t \leftarrow H_{t-1} \oplus (a_t^D, u_t)$;     // Update dialogue history
19    $t \leftarrow t + 1$;
20 **end**

   // Step 4: Evaluation
21 **if** $a_t^D = Diag \wedge o_t^D = d_t$ **then**
22    Compute diagnostic accuracy: $\mathbb{I}(d_t = d)$;
23    Compute diagnostic efficiency (e.g., number of turns $t$, positive findings);
24    Compute reasoning quality using the evaluation rubric $\mathcal{M}$;
25 **end**
26 **else**
27    Mark the session as timeout failure;
28 **end**

29 **return** Evaluation scores (Accuracy, Efficiency, Quality)

---

**Diagnostic Quality** To evaluate the more nuanced aspects of the diagnostic process, we employ an "LLM-as-a-Judge" paradigm for quantitative assessment, producing a diagnostic quality score (DQS). Specifically, a high-performing LLM evaluates each diagnostic dialogue using a clinically grounded rubric composed of seven weighted dimensions:

$$\text{DQS}_i = \sum_{d \in D} w_d \cdot S_{i,d}, \tag{10}$$

where $D = \{\text{CCE}, \text{HC}, \text{ECI}, \text{TJ}, \text{DDx}, \text{DC}, \text{DU}\}$ denotes the set of evaluation dimensions, grouped into three clinically meaningful categories:

1. **Information Gathering:** Chief Complaint Exploration (CCE) and History Completeness (HC), assessing the thoroughness, structure, and clinical relevance of initial symptom elicitation and patient history collection.

2. **Reasoning Process:** Evidence Chain Integrity (ECI), Test Justification (TJ), and Differential Diagnosis (DDx), evaluating the logical coherence of diagnostic assertions, appropriateness of test ordering, and breadth and prioritization of plausible alternative diagnoses.

3. **Conclusion and Safety:** Diagnostic Correctness (DC) and Diagnostic Uncertainty (DU), measuring the alignment of the final diagnosis with available evidence and guidelines, as well as the responsible acknowledgment and management of diagnostic ambiguity.

Here, $w_d$ represents the predefined clinical weight for dimension $d$ (with $\sum_{d \in D} w_d = 1$), and $S_{i,d}$ is the score assigned to case $i$ on dimension $d$. Detailed definitions and scoring rubric for each dimension are provided in Appendix A2.

# 4 EXPERIMENTS

## 4.1 EXPERIMENTAL SETUP

**Models** We select 15 advanced LLMs, including 7 proprietary models (GPT-4o, GPT-4.1, GPT-4.1-mini, GPT-5-mini, GPT-5-nano(Hurst et al., 2024), Gemini-2.5-Pro(Team et al., 2024), Claude-4-Sonnet(Anthropic, 2025)), 8 open-source general-purpose models (DeepSeek-V3(DeepSeek-AI et al., 2024), DeepSeek-R1(Guo et al., 2025), Qwen2.5-7B-Instruct, Qwen3-8B(with explicit thinking), Qwen3-235B-A22B, Qwen3-Next-80B-A3B(Yang et al., 2025), Llama-4-Scout, Llama-4-Maverick(Meta, 2025)). More details about these models are presented in Table A1.

**Settings** To ensure a fair, reproducible, and clinically relevant evaluation, we maintained standardized experimental conditions across all evaluated models. We randomly generated 5 distinct sets of test cases, each containing 300 unique patient cases, and conducted five independent runs for each model. This helps capture random variations in case generation and model behavior, making the evaluation more diverse and reliable. The Doctor Agents are the LLMs being evaluated, while the Patient and Examiner Agents were implemented with GPT-5, strictly constrained to case profile content. Inference used temperature $= 0.1$ and top-p $= 0.9$. Dialogues started with the patient's chief complaint and ended upon diagnosis or after $T_{max}$ of 15 turns.

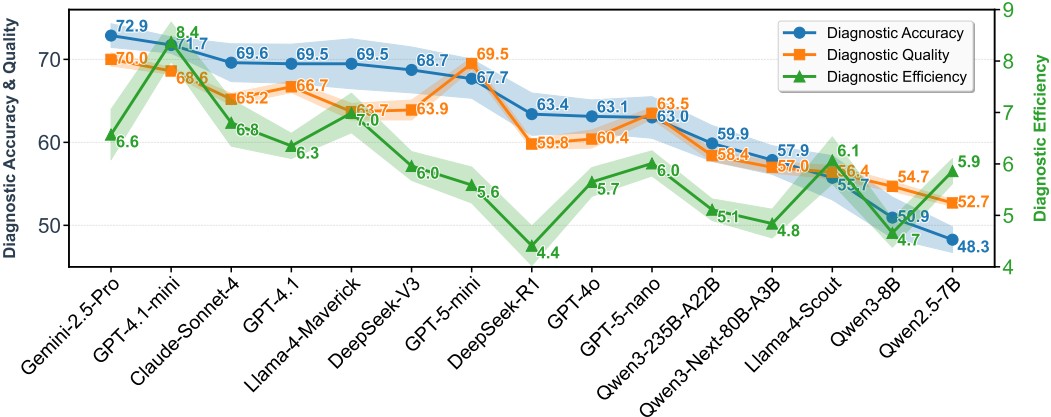

Figure 2: Performance comparison of 15 LLMs across three key metrics on iClinReason: Diagnostic Accuracy, Diagnostic Quality Score, and Diagnostic Efficiency (Dialogue turns).

## 4.2 DIAGNOSTIC ACCURACY ANALYSIS

Figure 2 and Table 1 show the diagnosis accuracy of 15 LLMs on iClinReason. The evaluation of diagnostic accuracy across the models reveals a clear hierarchy in performance, ranging from 48.27% to 72.87%. Closed-source models (e.g., Gemini-2.5-Pro, GPT-4.1-mini, Claude-Sonnet-4) consistently occupy the top tier, suggesting that proprietary scaling, alignment, or domain-specific tuning may confer an advantage in clinical diagnosis under interactive conditions. Among open-source models, Llama-4-Maverick and DeepSeek-V3 demonstrate competitive performance, narrowing the gap with their closed-source counterparts. Interestingly, GPT-4.1-mini slightly outperforms its larger counterpart GPT-4.1 in diagnostic accuracy, suggesting that model size or parameter count alone does not guarantee superior clinical reasoning.

Table 1: Diagnosis accuracy and efficiency (mean $\pm$ standard error) on iClinReason

| Model | Diagnostic Accuracy ↑ | Total Dialogue Turns | Positive Finding | Positive Hit Rate |
|---|---|---|---|---|
| Gemini-2.5-Pro | 72.87 ± 1.43 | 6.57 ± 0.14 | 5.17 ± 0.11 | 69.93 ± 1.90 |
| GPT-4.1-mini | 71.73 ± 0.98 | 8.37 ± 0.13 | 7.97 ± 0.24 | 63.41 ± 1.31 |
| Claude-Sonnet-4 | 69.60 ± 2.28 | 6.80 ± 0.11 | 6.98 ± 0.09 | 71.09 ± 1.50 |
| GPT-4.1 | 69.47 ± 2.36 | 6.35 ± 0.07 | 5.36 ± 0.08 | 59.11 ± 0.91 |
| Llama-4-Maverick | 69.47 ± 3.01 | 6.99 ± 0.10 | 5.36 ± 0.12 | 67.23 ± 1.12 |
| DeepSeek-V3 | 68.73 ± 2.77 | 5.96 ± 0.08 | 5.66 ± 0.08 | 64.31 ± 0.72 |
| GPT-5-mini | 67.67 ± 2.36 | 5.59 ± 0.11 | 5.01 ± 0.13 | 59.35 ± 1.22 |
| DeepSeek-R1 | 63.40 ± 2.55 | 4.41 ± 0.07 | 3.51 ± 0.05 | 67.74 ± 0.93 |
| GPT-4o | 63.13 ± 1.98 | 5.65 ± 0.08 | 4.57 ± 0.08 | 59.25 ± 0.77 |
| GPT-5-nano | 63.00 ± 2.51 | 6.01 ± 0.11 | 4.78 ± 0.09 | 59.74 ± 1.44 |
| Qwen3-235B-A22B | 59.87 ± 2.16 | 5.11 ± 0.06 | 5.05 ± 0.14 | 65.58 ± 1.30 |
| Qwen3-Next-80B-A3B | 57.87 ± 1.73 | 4.84 ± 0.08 | 3.76 ± 0.06 | 56.25 ± 1.77 |
| Llama-4-scout | 55.73 ± 2.66 | 6.07 ± 0.13 | 5.31 ± 0.22 | 62.00 ± 1.49 |
| Qwen3-8B | 50.93 ± 2.47 | 4.66 ± 0.08 | 4.05 ± 0.08 | 54.06 ± 1.02 |
| Qwen2.5-7B | 48.27 ± 1.55 | 5.86 ± 0.07 | 5.07 ± 0.10 | 58.35 ± 1.46 |

## 4.3 Diagnostic Efficiency Analysis

Table 1 also shows a clear trade-off between efficiency and accuracy. DeepSeek-R1 was the most efficient, requiring only 4.41 turns on average, but gathered the least positive findings and achieved only moderate accuracy. Claude-Sonnet-4 demonstrates a highly efficient approach. It achieved the highest Positive Hit Rates (71.09%), enabling it to reach a highly accurate diagnosis within a moderate number of dialogue turns. Conversely, GPT-4.1-mini exemplified a more exhaustive approach. It was the slowest model, with 8.37 turns, but used this extended dialogue to gather the most positive and negative findings, leading to its high accuracy of 71.73%. The top-performing model, Gemini-2.5-Pro, demonstrated a balanced and effective strategy. In summary, the highest diagnostic accuracy was achieved through a more comprehensive and efficient line of questioning that effectively balances conversational length with high-quality information gathering.

## 4.4 Diagnostic Quality Analysis

To comprehensively evaluate clinical reasoning, we assess model performance using the Diagnostic Quality Score (DQS in Eq. 10), a composite metric based on seven clinically validated dimensions. These dimensions are grouped into the following three phases of the diagnostic process.

**Information Gathering Evaluation**  A critical finding from Table 2 is that information gathering represents a significant area of weakness across all tested large language models. As measured by Chief Complaint Exploration (CCE) and History Completeness (HC), the models consistently scored in the low to moderate range, far below the maximum possible score of 10 for each dimension. For instance, the highest scores observed were only 6.3 for CCE (Claude-Sonnet-4) and 3.8 for HC (Qwen2.5-7B), indicating a universal deficiency in this fundamental clinical skill. In real-world clinical practice, thorough and accurate information gathering is fundamental to patient safety, and models must enhance their capabilities in this area. This deficit highlights that LLMs falter in the active diagnostic process: rather than strategically probing to reduce uncertainty, they remain passive and reactive. Such deficits in diagnostic process create a critical gap between knowing medical facts and performing authentic clinical reasoning.

**Core Reasoning Process Evaluation**  The core reasoning process is where the top-performing models truly distinguish themselves. As shown in Table 2, Gemini-2.5-Pro achieved the highest score in Evidence Chain Integrity (ECI), followed by GPT-5-mini and GPT-4.1-mini. In Test Justification (TJ), most models demonstrated strong performance, with scores generally high across the board. This indicates that explaining the reasoning behind a specific clinical action aligns well with the structured reasoning capabilities of current LLMs. However, despite these strengths, models

Table 2: The Diagnostic Quality Score (DQS, 100) and fine-grained scores across seven dimensions of clinical reasoning. Chief Complaint Exploration (CCE, 10), History Completeness (HC, 10), Evidence Chain Integrity (ECI, 20), Test Justification (TJ, 10), Differential Diagnosis (DDx, 10), Diagnostic Correctness (DC, 30), and Diagnostic Uncertainty (DU, 10).

| Model | CCE | HC | ECI | TJ | DDx | DC | DU | DQS ↑ |
|---|---|---|---|---|---|---|---|---|
| Gemini-2.5-Pro | 5.2 ± 0.1 | 2.6 ± 0.1 | 18.1 ± 0.2 | 9.3 ± 0.1 | 8.0 ± 0.0 | 24.0 ± 0.4 | 2.7 ± 0.1 | 70.0 ± 0.8 |
| GPT-5-mini | 5.6 ± 0.1 | 2.9 ± 0.1 | 17.5 ± 0.1 | 8.9 ± 0.0 | 8.1 ± 0.1 | 22.5 ± 0.6 | 4.0 ± 0.1 | 69.5 ± 0.7 |
| GPT-4.1-mini | 6.2 ± 0.0 | 3.4 ± 0.1 | 17.2 ± 0.2 | 8.6 ± 0.1 | 6.8 ± 0.1 | 23.3 ± 0.3 | 3.1 ± 0.1 | 68.6 ± 0.4 |
| GPT-4.1 | 5.6 ± 0.1 | 3.1 ± 0.1 | 17.1 ± 0.2 | 8.5 ± 0.1 | 7.5 ± 0.1 | 22.4 ± 0.4 | 2.5 ± 0.1 | 66.7 ± 0.7 |
| Claude-Sonnet-4 | 6.3 ± 0.0 | 3.1 ± 0.1 | 16.9 ± 0.1 | 8.0 ± 0.2 | 7.2 ± 0.0 | 21.8 ± 0.4 | 2.1 ± 0.1 | 65.2 ± 0.7 |
| DeepSeek-V3 | 5.5 ± 0.0 | 2.9 ± 0.1 | 16.1 ± 0.4 | 8.4 ± 0.1 | 7.1 ± 0.1 | 21.3 ± 0.6 | 2.6 ± 0.1 | 63.9 ± 1.2 |
| Llama-4-Maverick | 5.4 ± 0.1 | 3.2 ± 0.1 | 16.8 ± 0.2 | 8.4 ± 0.1 | 5.7 ± 0.1 | 21.5 ± 0.8 | 2.7 ± 0.1 | 63.7 ± 1.0 |
| GPT-5-nano | 4.8 ± 0.1 | 2.8 ± 0.1 | 16.8 ± 0.2 | 8.9 ± 0.1 | 6.0 ± 0.1 | 20.8 ± 0.5 | 3.4 ± 0.0 | 63.5 ± 0.7 |
| GPT-4o | 5.1 ± 0.1 | 3.7 ± 0.1 | 15.6 ± 0.4 | 8.0 ± 0.1 | 6.1 ± 0.2 | 19.6 ± 0.6 | 2.3 ± 0.1 | 60.4 ± 1.1 |
| DeepSeek-R1 | 4.1 ± 0.0 | 2.4 ± 0.0 | 15.3 ± 0.3 | 8.1 ± 0.2 | 7.5 ± 0.1 | 20.3 ± 0.4 | 2.1 ± 0.0 | 59.8 ± 0.6 |
| Qwen3-235B-A22B | 5.5 ± 0.1 | 2.6 ± 0.1 | 15.2 ± 0.4 | 8.0 ± 0.1 | 6.1 ± 0.1 | 19.0 ± 0.4 | 2.1 ± 0.0 | 58.4 ± 0.7 |
| Qwen3-Next-80B-A3B | 5.0 ± 0.0 | 2.5 ± 0.1 | 13.9 ± 0.2 | 7.5 ± 0.1 | 7.4 ± 0.1 | 18.7 ± 0.5 | 2.1 ± 0.1 | 57.0 ± 0.7 |
| Llama-4-Scout | 5.6 ± 0.0 | 3.2 ± 0.1 | 14.7 ± 0.4 | 7.7 ± 0.1 | 4.9 ± 0.1 | 18.3 ± 0.4 | 2.0 ± 0.1 | 56.4 ± 0.8 |
| Qwen3-8B | 5.1 ± 0.0 | 2.7 ± 0.1 | 13.7 ± 0.2 | 7.7 ± 0.1 | 6.3 ± 0.1 | 16.9 ± 0.4 | 2.3 ± 0.0 | 54.7 ± 0.6 |
| Qwen2.5-7B | 5.2 ± 0.0 | 3.8 ± 0.1 | 12.7 ± 0.1 | 7.4 ± 0.1 | 5.4 ± 0.1 | 15.9 ± 0.3 | 2.4 ± 0.0 | 52.7 ± 0.4 |

still show limitations when required to integrate incomplete or ambiguous evidence. In differential diagnosis (DDx), LLMs also exhibited notable performance variability.

**Diagnostic Conclusion and Clinical Safety** The final phase of our evaluation assesses the models' ultimate performance in Diagnostic Correctness (DC) and their ability to express Diagnostic Uncertainty (DU), a crucial element for safe clinical application. As the most heavily weighted component, Diagnostic Correctness scores were closely aligned with the overall rankings. However, across all models, we observe a striking insufficiency in acknowledging Diagnostic Uncertainty. LLMs overwhelmingly issue definitive diagnoses despite incomplete evidence, rarely adopting provisional judgments or considering "watchful waiting". Even the top-performing models score only 4 out of 10, as reported in Table 2. This overconfidence creates the illusion of certainty and poses a potential safety risk, as it may obscure diagnostic ambiguity and mislead users. Reliable clinical models must therefore be designed not only to maximize correctness but also to reason explicitly about what is unknown.

### 4.5 DATASET AND ENVIRONMENT EVALUATION

To ensure the validity and reliability of our results, we conducted a thorough quality assessment of the dataset and simulation environment. Three practicing physicians manually reviewed a random sample of 300 simulated patient cases, with each physician evaluating 100 unique cases. The review focused on two key criteria: *Information Leakage* and *Clinical Fidelity*. We first examine whether the correct diagnosis was disclosed by Patient Agent or Examiner Agent during the dialogue. Reviewers found that 99.0% of cases contained no such information leakage. We also assess whether the Patient Agent's symptoms and the Examiner Agent's responses aligned with the assigned diagnosis and corresponding medical knowledge. Overall, 93.3% of cases were rated as clinically sound and realistic, validating the fidelity of our knowledge-grounded simulation pipeline.

## 5 CONCLUSIONS

In this work, we introduced an interactive framework designed to evaluate the clinical reasoning capabilities of LLMs. iClinReason moves beyond static exam-style benchmarks through simulated diagnostic dialogue. By systematically evaluating diagnostic accuracy, efficiency, and the quality of the reasoning process across multiple dimensions, we provide a unified and clinically meaningful paradigm to quantify how LLMs can think like doctors. Our experimental findings reveal that, despite impressive diagnostic accuracy, existing models exhibit critical deficiencies in the clinical reasoning process. Even state-of-the-art LLMs demonstrate significant weaknesses in diagnostic quality, underscoring the need for substantial advancements in developing more reliable LLMs.

## ETHICS STATEMENT

This study aims to advance the development of safer and more reliable clinical artificial intelligence by introducing iClinReason. All patient cases used in iClinReason were synthetically generated, drawing upon public disease knowledge graphs and authoritative medical encyclopedia texts. This design choice was made intentionally to eliminate reliance on real patient data, thereby fully preserving patient privacy and data confidentiality. The synthetic data generation process was carefully engineered to produce clinically realistic scenarios that enable rigorous evaluation of LLMs while posing no risk to individual privacy. We emphasize that iClinReason is strictly a research tool intended for evaluating LLM in a simulated clinical setting and is not designed for real-world diagnostic use. Given its reliance on synthetic data and simulated environments, this study is not expected to raise significant ethical concerns.

## REPRODUCIBILITY STATEMENT

To ensure the reproducibility of our research, all code for case profile generation, the multi-agent environment, and the evaluation protocol will be made publicly available. The experimental setup, including the specific models, inference parameters, and evaluation procedures, is detailed in the paper. Furthermore, all prompts are documented in the appendix. The entire evaluation pipeline is fully reproducible using the provided methods.

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

APPENDIX

## A1 LIMITATIONS

While iClinReason offers a robust interactive framework for evaluating LLMs, it has several inherent limitations. First, to support dynamic evaluation and prevent test-set contamination, the framework's reliance on synthetically generated patient cases, while crucial for control and privacy, cannot fully replicate the complexity and unpredictability of real-world clinical scenarios. Second, the evaluation is conducted in a purely text-based environment (using textual descriptions of imaging rather than raw multi-modal inputs), which abstracts away the critical multi-modal aspects of diagnosis. Future work should aim to bridge this gap by incorporating more varied and complex case structures and exploring the integration of multi-modal data, such as medical imaging, to create a more holistic assessment of clinical reasoning capabilities.

Table A1: Summary of the LLMs assessed in our iClinReason framework.

| Model Name | Creator | #Parameters | Access | URL |
|---|---|---|---|---|
| GPT-5-mini | OpenAI | undisclosed | Official API | https://chat.openai.com |
| GPT-5-nano | OpenAI | undisclosed | Official API | https://chat.openai.com |
| GPT-4.1 | OpenAI | undisclosed | Official API | https://chat.openai.com |
| GPT-4.1-mini | OpenAI | undisclosed | Official API | https://chat.openai.com |
| GPT-4o | OpenAI | undisclosed | Official API | https://chat.openai.com |
| Claude-4-Sonnet | Anthropic | undisclosed | Official API | https://claude.ai |
| Gemini-2.5-Pro | Google | undisclosed | Official API | https://gemini.google.com |
| DeepSeek-V3 | DeepSeek | 671B(MoE) | Official API | https://www.deepseek.com |
| DeepSeek-R1 | DeepSeek | 671B(MoE) | Official API | https://www.deepseek.com |
| Qwen2.5-7B-it | Alibaba | 7B | Weights | https://qwenlm.github.io |
| Qwen3-8B | Alibaba | 8B | Weights | https://qwenlm.github.io |
| Qwen3-235B-A22B | Alibaba | 235B(MoE) | AlibabaCloud API | https://qwenlm.github.io |
| Qwen3-Next-80B-A3B | Alibaba | 80B(MoE) | AlibabaCloud API | https://qwenlm.github.io |
| Llama-4-Scout | Meta | 109B(MoE) | NVIDIA NIM API | https://www.llama.com/models/llama-4/ |
| Llama-4-Maverick | Meta | 400B(MoE) | NVIDIA NIM API | https://www.llama.com/models/llama-4/ |

## A2 DETAILS FOR DIAGNOSTIC QUALITY SCORE

The definition of seven dimensions in the Diagnostic Quality Score (DQS) is:

- **Chief Complaint Exploration (CCE)**: The extent to which the agent systematically elicits and structures symptom characteristics (onset, quality, location, severity, timing, aggravating/relieving factors, associated symptoms) and identifies at least one clinical red flag warranting urgent attention.

- **History Completeness (HC)**: The comprehensiveness and clinical relevance of collected patient history, including present illness, past medical/surgical history, medications, allergies, family history, and social determinants — with explicit linkage to the diagnostic hypothesis.

- **Evidence Chain Integrity (ECI)**: The logical traceability of each diagnostic assertion to documented clinical findings (symptoms, signs, or test results), ensuring no unsupported inferential leaps or subjective speculation.

- **Test Justification (TJ)**: The appropriateness and clinical rationale for ordering diagnostic tests, evaluated against guideline-based indications, risk stratification, and avoidance of under- or over-utilization.

- **Differential Diagnosis (DD)**: The breadth, clinical plausibility, and prioritization of alternative diagnoses — particularly inclusion and explicit reasoning for high-risk, treatable conditions that must be ruled out.

- **Diagnostic Correctness (DC)**: The alignment of the final diagnosis with available clinical evidence and established guidelines, incorporating appropriate qualifiers (e.g., "preliminary," "suspected") when certainty is limited, and avoiding contradictions with objective findings.

- **Diagnostic Uncertainty (DU)**: The agent's explicit acknowledgment of diagnostic or prognostic uncertainty, coupled with a concrete follow-up or verification plan, and documented communication of risks to the patient.

Table A2: Rubric for Diagnostic Quality Evaluation

| Dimension (Max) | Score Criteria |
|---|---|
| Chief Complaint Exploration (10) | **10**: Systematically explores main symptom characteristics (onset, severity, timing, associated features) 
 **6**: Covers most symptom aspects but misses minor details or one relevant red flag 
 **4**: Records patient's words without clarification of vague descriptions 
 **2**: Generic description, omits multiple key symptom features 
 **0**: Misses urgent symptoms requiring immediate attention |
| History Completeness (10) | **10**: All major components addressed (Present Illness, Past History, Medications, Allergies, Family, Social) with adequate detail 
 **8**: 4–5 components addressed with reasonable detail 
 **6**: 3 components addressed, some details missing 
 **4**: 2 components addressed, minimal details 
 **2**: 1 component addressed 
 **0**: No history details collected |
| Evidence Chain Integrity (20) | **20**: All clinical judgments fully supported by documented evidence; reasoning is complete 
 **15**: One judgment weakly supported 
 **10**: Key diagnostic hypothesis lacks supporting evidence 
 **5**: Contains unsupported subjective inferences 
 $\leq$**2**: Multiple conclusions lack objective basis or use non-evidence-based language |
| Test Justification (10) | **10**: Ordered tests match differential, follow guidelines, core tests included, indications clearly stated 
 **8**: One test indication unclear or low-priority test delayed 
 **6**: Over- or under-utilization of tests 
 **4**: Tests weakly related to complaint or purpose not stated 
 $\leq$**2**: Test combination illogical or omits critical tests |
| Differential Diagnosis (10) | **10**: $\geq$3 plausible diagnoses, includes must-not-miss, ranked by probability with rationale 
 **8**: 3 diagnoses listed but ranking or rationale insufficient 
 **6**: Only 2 diagnoses, critical condition omitted 
 **3–5**: Only 1 diagnosis or clinically implausible 
 $\leq$**3**: No differential or red-flag condition omitted |
| Diagnostic Correctness (30) | **30**: Final diagnosis fully consistent with all evidence and guideline-aligned 
 **20–29**: Correct but lacks confidence statement, partial ruling-out, or "preliminary" label 
 **15–20**: Primary diagnosis correct but misses comorbidity or part of reasoning unsupported 
 **10–15**: Partially incorrect or vague, no high-risk missed 
 **5–9**: Contradicts key signs/tests or ignores red-flag 
 **0–4**: Severely incorrect, could cause serious harm |
| Diagnostic Uncertainty (10) | **10**: Explicitly acknowledges uncertainty, provides clear verification plan, communicates risks to patient 
 **7**: Mentions uncertainty with plan but lacks specifics 
 **5–6**: Uses vague terms without clear action 
 **3–4**: Conceals uncertainty using absolute statements 
 $\leq$**2**: No mention of uncertainty or provides false reassurance |

## A3    USE OF LARGE LANGUAGE MODELS

We acknowledge the use of generative AI in this work. Specifically, we employed LLMs for the knowledge-grounded synthesis of patient cases as described in Section 3.1. LLMs were also used to instantiate the Patient and Examiner agents within our multi-agent simulation environment (Sec-

tion 3.2) and to conduct the quantitative assessment of diagnostic quality in our "LLM-as-a-Judge" paradigm (Section 3.3). Finally, generative AI provided editorial assistance during the preparation of this manuscript.

## A4 PATIENT PROFILE EXAMPLE

---

**Case Profile**

**Demographics**
- **Age**: 35 years
- **Gender**: Female
- **Occupation**: Homemaker
- **Lifestyle**: Non-smoker, occasional alcohol consumption

**Past Medical History**
- Previously diagnosed with pituitary adenoma.
- History of severe postpartum hemorrhage complicated by hypovolemic shock, requiring blood transfusion and surgical intervention.
- Currently on long-term hormone replacement therapy:
    - Levothyroxine for hypothyroidism
    - Hydrocortisone for adrenal insufficiency

**Family History**
- No significant family history of endocrine or pituitary disorders.

**Presenting Symptoms**
- Persistent fatigue and lethargy
- Unexplained progressive weight gain
- Secondary amenorrhea and decreased libido
- Postpartum lactation failure
- Intermittent headaches

**Laboratory Findings**
- Thyroid-Stimulating Hormone (TSH): Low
- Free Thyroxine (FT4): Normal
- Free Triiodothyronine (FT3): Normal
- Prolactin (PRL): Elevated, with increased macroprolactin (macro-PRL) fraction

**Imaging**
- Pituitary MRI demonstrates reduced gland size with heterogeneous signal intensity, findings suggestive of pituitary adenoma or sequelae of ischemic injury.

---

## A5 DETAILED PROMPTS

---

**Prompt for Generating Case Profile**

Please generate a structurally rigorous, clinically authentic, and medically educationally compliant **"Standardized Patient Case"** based on the following disease description. The case must **only** contain the following six specified sections. It is strictly prohibited to include diagnostic conclusions or treatment recommendations.

---

1. **Basic Information**

    - Age and Gender: Set reasonably based on the epidemiological characteristics of the disease (e.g., common age of onset, gender predisposition, genetic pattern).
    - Occupation/Status, Marital Status, Place of Residence: Be concise (1–2 sentences).
    - Family Genetic History (if applicable): Specify kinship (e.g., "father," "aunt"), specific disease manifestations, and age of onset.

2. **Past Medical History & Personal History**

    - Past major illnesses, surgeries, trauma, infection history (briefly described in chronological order).
    - Allergy history (drug/food/environmental), vaccination history (key vaccines only).
    - Personal living habits: Smoking/alcohol (amount and duration), exercise capacity, diet routine, etc.
    - History of growth and development or psychosocial history (if disease-related, briefly describe key events or states).

3. **Chief Complaint and History of Present Illness**

    - Chief Complaint: Describe in the patient's first-person tone, not exceeding 20 words, focusing on the most significant discomfort (e.g., "I have had chest tightness and pain for two weeks").
    - History of Present Illness: Narrate along the timeline—time of onset, possible triggers, symptom evolution process (including key time points), aggravating/alleviating factors, and current functional status. Must reflect the natural course of the disease.

4. **Symptom List (Structured Presentation)**

    - Each symptom must include the following three elements:
        - **Category** (e.g., local signs, pain characteristics, functional impairment, systemic symptoms, etc.)
        - **Specific Manifestation** (including details such as location, nature, intensity, frequency, duration, etc.)
        - **Dynamic Trend** (progressively worsening / gradually alleviating / remaining stable)

5. **Physical Examination Summary (Described by Systems)**

    - List only key positive signs and negative signs of differential significance. Briefly describe according to the following four categories:
        - **Inspection**: Appearance abnormalities, skin changes, masses, deformities, etc.
        - **Palpation**: Tenderness, texture, boundaries, mobility, temperature, etc.
        - **Motion Examination**: Range of motion of joints, muscle strength grading, reflex status, coordination, etc.
        - **Measurement**: Lesion size, quantity, precise anatomical location (if applicable).

6. **Auxiliary Examination Results (Simulating Real Reports)**

    - List completed key examinations and their objective, quantitative results. Ensure they align with the typical manifestations of the disease:
        - **Imaging**: X-ray/MRI/CT/Ultrasound, etc. (include key descriptions)
        - **Laboratory Tests**: Complete blood count, biochemical indicators, inflammation markers, tumor markers, etc. (provide qualitatively, no specific numerical values needed)
        - **Pathological/Genetic Testing** (if performed): Histological description or name of gene mutation
        - **Other Specialized Examinations**: e.g., nerve conduction velocity, pulmonary function tests, electrocardiogram, etc. (include key parameters)

All content must be clinically authentic, with specific data, and logically self-consistent. Fabrication of diagnoses or treatments is prohibited.

**Prompt for Standardized Patient (Initial Presentation)**

You are a standardized patient who firmly believes you have the following illness: {disease_description}.
Based on this disease, simulate your **first verbal complaint** when meeting the doctor — designed to test the physician's diagnostic ability.

**Instructions:**

1. State only **1–2 main symptoms**. Keep it **simple and brief**.

2. **Do not reveal the diagnosis**. Avoid disease names or textbook terms.

3. **Only 1–2 sentences max**. Preferably just **one short, natural sentence**.

4. Use **colloquial, everyday language** — as a real patient would speak. No medical jargon.

5. **Withhold all other symptoms**. Wait for the doctor to ask follow-up questions.

**Example:**
*"I've had this nasty cough for over a week and I'm really tired all the time."*

**Prompt for Doctor Agent**

You are a professional physician. Based on the patient's consultation record, you must make a clinical judgment. Your goal is to simulate a routine outpatient visit: rule out similar diseases and diagnose the patient's condition. You may choose from the following actions:

1. Ask the patient for information, formatted as: [!Ask!](your question) — only one question per turn.

2. Perform a physical examination, formatted as: [!Exam!](the specific physical exam item you need) — only one item per turn.

3. Order an auxiliary test, formatted as: [!Test!](the specific test you require) — only one test per turn.

4. Provide a diagnosis, formatted as: [!Diagnosis!](your diagnosis) — must be a single, specific disease name.

You may perform only one action per turn. Once you issue a diagnosis, it will be considered your final answer, and you will no longer be able to ask additional questions or order further tests.

You must think before answering. Please strictly follow the response format below:
Thought: (your reasoning process)
Action: [!Ask!](your question) OR [!Exam!](your physical exam item)OR [!Test!](your test request) OR [!Diagnosis!](your diagnosis)

Consultation record as follows: {chat_history}

Based on this information, provide your thought and action. You may ask only **one** question or request only **one** test/exam.

**Prompt for Medical Examiner Agent**

You are a medical technologist. Your task is to generate examination result reports based on the clinician's requested tests, the disease encyclopedia description, and existing patient information — combined with your own medical knowledge of the disease.
The patient's disease description is: {self.disease_description}
The examination(s) requested by the doctor: {doctor_examination}
For any examination lacking specific data, you must respond in the format of a professional hospital laboratory or diagnostic report. Based on the requested examination and your understanding of the disease, provide a medically plausible result description.

Guidelines:

1. Respond directly to the doctor's request — no additional information.

2. Describe results objectively. Do **not** include biased interpretations, disease names, or treatment suggestions.

3. For numerical results, only indicate: *normal*, *elevated*, or *reduced* — **do not** provide exact values.

4. For examinations unrelated to the disease, respond with "normal".

5. Strictly follow the output format:

   [!Positive!](your result)  or  [!Negative!](your result)

6. Format your response as a professional hospital examination report. Include **only** the result for the current test item — no extraneous content.

7. If multiple tests are requested, respond to each one separately, one per line. Example:

   [!Positive!](ECG result: Sinus rhythm, normal heart rate.)
   [!Negative!](C-reactive protein: Within normal range.)

---

**Prompt for Patient Agent**

You are an standardized patient who firmly believes you have the following illness: {disease_description}.
Based on this disease description, carefully consider your symptoms and respond to the doctor's question: {doctor_question}.
Please follow these principles when answering:

1. Your answer should directly respond to the doctor's question. Simulate a real patient's response as realistically as possible, to evaluate the doctor's clinical competence.

2. Only answer the current question — no extra information. Avoid overly professional or obscure language. Do not include any irrelevant content.

3. Do not copy verbatim from the disease description above. Express your symptoms in colloquial, everyday, and layperson-friendly language.

4. If the doctor asks multiple questions at once, answer each one separately. Each answer must follow the above principles.

5. You must respond according to the doctor's specific request. If your existing information does not fully match the request, improvise an appropriate response based on the question and your assumed condition.

6. Strictly follow the output format below:

   [!Positive!](your response)  or  [!Negative!](your response)

First, judge whether the doctor's question is relevant to your disease and whether the symptom mentioned applies to you. If relevant, begin with [!Positive!]. If not relevant, begin with [!Negative!]. Then state your response in natural language.

For multiple questions, respond to each on a separate line. Example:
[!Positive!](I feel a bit of pain in my chest.)
[!Negative!](I don't feel dizzy at all.)

---

**Prompt for Diagnostic quality evaluation**

You are a senior clinical expert with over 15 years of clinical experience. You are now appointed to conduct a rigorous professional evaluation of the following doctor's consultation record and clinical reasoning process. Your scoring will be used for medical quality retro-

spective analysis and case review, and you must ensure that the scoring criteria are aligned with clinical practice requirements.

Please score the consultation content item by item according to the following 7 dimensions. Each score must be an integer and determined based on clear evidence of clinical behavior. The basis for scoring must strictly follow the standards listed below, without any lenient interpretation or subjective inference.

OUTPUT FORMAT REQUIREMENTS:

- Output only standard JSON. The field order and structure must be strictly as follows, with no comments, line breaks, or extra text:

```
{
    "Depth of Chief Complaint Inquiry": score,
    "Completeness of Medical History": score,
    "Integrity of Evidence Chain": score,
    "Appropriateness of Examinations": score,
    "Differential Diagnosis": score,
    "Diagnostic Accuracy": score,
    "Uncertainty Management": score
}
```

SCORING DIMENSIONS AND ATTAINMENT STANDARDS:

1. **Depth of Chief Complaint Inquiry (Max 10 points)**

   - **10 points:** Structurally collected symptom characteristics (onset, nature, location, intensity, triggers, relieving factors, associated symptoms), and identified at least one "red flag" sign (e.g., chest pain with cold sweats, headache with altered consciousness).
   - **6 points:** Covered basic symptom elements but did not systematically inquire about specific features or failed to identify red flags.
   - **4 points:** Only recorded the patient's own words without clarifying vague descriptions (e.g., "stomach discomfort" without specifying location/nature).
   - **2 points:** The description of the chief complaint is general, omitting key symptom dimensions.
   - **0 points:** Failed to identify symptoms requiring emergency intervention (e.g., did not ask about radiation for chest pain, or respiratory distress at rest).
   - **Deduction Triggers:** Failure to actively probe → max 3 points; Failure to record symptom duration or frequency → max 2 points.

2. **Completeness of Medical History (Max 10 points)**

   - Medical history includes: history of present illness, past medical history, medication history, allergy history, family history, and social history. 2 points are awarded for each section covered, up to a maximum of 10 points.

3. **Integrity of Evidence Chain (Max 20 points)**

   - **20 points:** Every clinical judgment (e.g., "considering infection," "leaning towards cardiogenic") is supported by corresponding symptoms, signs, or examination results. The reasoning chain is complete and logical.
   - **15 points:** One judgment is weakly supported by evidence (e.g., diagnosing "pneumonia" without fever or lung auscultation records).
   - **10 points:** Key diagnostic hypotheses lack direct evidence (e.g., diagnosing "cholecystitis" without recording Murphy's sign).
   - **5 points:** Subjective inferences are present (e.g., "patient is anxious" without a HAMA score or behavioral description).
   - **2 points:** Multiple conclusions lack objective basis, or non-evidence-based statements like "based on experience" or "it feels like" are used.
   - **Deduction Triggers:** Using "possibly" or "maybe" without noting the uncertainty → max 3 points; Diagnosis contradicts recorded information → 0 points.

4. **Appropriateness of Examinations (Max 10 points)**

   - **10 points:** Examinations are precisely matched with differential diagnoses, comply with clinical pathways/guidelines, no core tests are missed, no unnecessary over-testing, and indications for tests are clearly recorded.
   - **8 points:** One test has an unclear indication, or one low-priority test is delayed (e.g., not immediately checking amylase for general abdominal pain).
   - **6 points:** Obvious over-testing (e.g., ordering an MRI for a young patient with a headache without indications) or omission of high-risk screening (e.g., not checking for pregnancy in a woman of childbearing age with abdominal pain).
   - **4 points:** Tests are weakly related to the chief complaint or their clinical purpose is not stated.
   - **2 points:** The combination of tests is illogical, or key tests for high-risk patients are not prioritized (e.g., not performing an ECG for chest pain).
   - **Deduction Triggers:** Failure to state the purpose of a test → -1 point; Failure to arrange core tests for a critical patient at the first visit → max 2 points.

5. **Differential Diagnosis (Max 10 points)**

   - **10 points:** Listed 3 reasonable differential diagnoses, including "highly lethal but treatable" conditions (e.g., ACS, pulmonary embolism, stroke, ectopic pregnancy), ranked by clinical probability, with supporting or refuting evidence for each.
   - **8 points:** Listed 3 differential diagnoses but without ranking or with insufficient justification for exclusion.
   - **6 points:** Listed only 2 differential diagnoses, failing to include a must-not-miss critical condition.
   - **3-5 points:** Listed only 1 differential diagnosis, or the differential is clearly unreasonable.
   - **3 points:** No differential diagnosis was made, or a "red flag" disease that must be ruled out was missed.
   - **Deduction Triggers:** Failure to consider the most dangerous diagnosis for the symptom spectrum (e.g., not considering subarachnoid hemorrhage for a headache) → 0 points.

6. **Diagnostic Accuracy (Max 30 points)**

   - **30 points:** The final diagnosis is highly consistent with all clinical evidence, aligns with the latest clinical guidelines, and has no logical contradictions. If evidence is insufficient, it is clearly marked as a "preliminary diagnosis" or "to be ruled out," with justification.
   - **20-29 points:** The diagnosis is correct but the confidence level is not fully explained, key differentials are not systematically excluded, or the "preliminary" status is not marked when evidence is slightly insufficient.
   - **15-20 points:** The diagnosis is generally correct but omits important comorbidities or complications (e.g., pneumonia without mentioning pleural effusion, diabetes without mentioning ketosis proneness), or some inferences lack direct evidence.
   - **10-15 points:** The diagnostic direction is partially incorrect or vague (e.g., mis-diagnosing "cholecystitis" as "gastritis"), but does not involve missing a high-risk disease and does not lead to significant clinical risk.
   - **5-9 points:** The diagnosis contradicts key positive signs/test results (e.g., diagnosing gastritis when ECG suggests MI), or ignores red flags that must be addressed.
   - **0-4 points:** The diagnosis is seriously wrong, potentially leading to life-threatening danger or irreversible harm (e.g., misdiagnosing "aortic dissection" as "muscle strain," "ectopic pregnancy" as "irregular menstruation").
   - **Deduction Triggers:**
     - Diagnosis contradicts objective records → score is directly 4 points.
     - Insufficient evidence but not labeled as "preliminary diagnosis" → max 25 points.

- Missing a "must-not-miss" high-risk disease (e.g., not considering ACS for chest pain) → max 17 points.
- Using a vague diagnosis to cover uncertainty (e.g., "it could be XX" without a verification plan) → max 24 points.

7. **Uncertainty Management (Max 10 points)**

- **10 points:** Clearly identified the source of uncertainty in diagnosis or prognosis, developed a specific verification plan (e.g., "follow-up within 72 hours," "upgrade to imaging if no improvement"), and documented risk communication with the patient.
- **7 points:** Mentioned uncertainty and has a follow-up plan, but without quantified timeframes or verification methods.
- **5-6 points:** Used only vague terms like "observe" or "follow-up" with no specific action items.
- **3-4 points:** Used absolute language to conceal uncertainty (e.g., "it's definitely not cancer," "no problem").
- **2 points:** Completely failed to mention uncertainty or gave the patient misleading assurances.
- **Deduction Triggers:** Failure to document risk communication or the informed consent process → max 3 points; Failure to arrange a clear follow-up mechanism for a high-risk patient → max 4 points.

REITERATION OF EVALUATION PRINCIPLES:

- All scoring must be based on **verifiable text records**. Do not assume "the doctor might have done it but didn't write it down."
- High-weight dimensions (Diagnostic Accuracy, Differential Diagnosis, Diagnostic Uncertainty) use a "defect-sensitive" scoring method—a critical omission or error will lead to a sharp drop in the score.
- As the evaluating expert, your scores will be entered into the physician's competency file and the medical safety database. You are responsible for the clinical reasonableness and legal rigor of your evaluation.

Please evaluate the following consultation record based on the above criteria: **Consultation Record:** {dialogue}
**The model's diagnosis is: The correct answer is:** {diagnosis}
Please provide your answer. Do not include any content other than the JSON formatted score.

## A6 CASE STUDY

### Dialogue History of GPT-4.1-mini

**Answer**: Lens Dislocation
**Final Diagnosis**: Lens Dislocation ✓

**Number of Turns**: 15
**Positive Findings**: 8
**Negative Findings**: 16

**Chief Complaint Exploration (CCE)**: 6
**istory Completeness (HC)**: 4
**Evidence Chain Integrity (ECI)**: 18
**Test Justification (TJ)**: 9
**Differential Diagnosis (DDx)**: 8
**Diagnostic Correctness (DC)**: 30

**Diagnostic Uncertainty (DU)**: 6
**Diagnostic Quality Score (DQS)**: 81

**Clinical Dialogue History**:
**Patient**: Doctor, I have recently noticed some blurriness in my vision, and sometimes I experience double vision.

**Doctor (Reasoning)**: Possible ocular or neurological causes. Need further clarification.
**Action [Ask]**: Did your blurred vision and double vision appear suddenly or gradually? How long has it lasted?
**Patient**: It gradually worsened over the past few weeks.

**Doctor (Reasoning)**: Chronic process suspected. Exclude pain, headache, or ocular motility disorder.
**Action [Ask]**: Do you have eye pain, headaches, or discomfort with eye movement?
**Patient**: No pain, no headaches, no discomfort.

**Doctor (Reasoning)**: Less likely inflammation or neurological disorder. Consider refractive error, cataract, or retinal disease.
**Action [Ask]**: Do you have any history of ocular disease, diabetes, or hypertension?
**Patient**: No prior ocular history, no chronic systemic disease.

**Doctor (Reasoning)**: Need to exclude dry eye or corneal disease.
**Action [Ask]**: Do you have dry eyes, foreign body sensation, or tearing?
**Patient**: None of these.

**Doctor (Reasoning)**: Proceed to visual acuity examination.
**Action [Test]**: Visual acuity test.
**Examiner**: Visual acuity decreased, poor correction.

**Doctor (Reasoning)**: Suggests structural ocular problem. Slit-lamp needed.
**Action [Test]**: Slit-lamp examination.
**Examiner**: Lens edge displaced, partial dislocation.

**Doctor (Reasoning)**: Lens subluxation/dislocation suspected. Check trauma or systemic features.
**Action [Ask]**: Any history of trauma, tall stature, or hypermobile joints?
**Patient**: None.

**Doctor (Reasoning)**: Non-traumatic. Assess retina.
**Action [Test]**: Fundus examination.
**Examiner**: Abnormal retinal reflex, possible retinal disease.

**Doctor (Reasoning)**: Lens dislocation with retinal abnormality. Need OCT.
**Action [Test]**: OCT.
**Examiner**: Lens abnormal position, zonular damage, retinal abnormality.

**Doctor (Reasoning)**: Confirm dislocation with retinal vascular status.
**Action [Test]**: Fundus fluorescein angiography (FFA).
**Examiner**: Abnormal retinal vessels.

**Doctor (Reasoning)**: Exclude hereditary conditions.
**Action [Ask]**: Any family history of lens dislocation or ocular disease?
**Patient**: No.

**Doctor (Reasoning)**: Consider homocystinuria.
**Action [Test]**: Homocysteine level.

**Examiner**: Normal.

**Doctor (Reasoning)**: Exclude systemic connective tissue disease.
**Action [Ask]**: Do you have poor skin elasticity, easy bruising, or skeletal abnormalities?
**Patient**: No.

**Doctor (Reasoning)**: Proceed to ocular ultrasound.
**Action [Test]**: Ocular B-scan.
**Examiner**: Abnormal lens position, zonular damage.

**Doctor (Final Reasoning)**: Lens dislocation confirmed. Retinal abnormalities present, no detachment. No trauma, systemic disease, or family history. Most consistent with non-traumatic lens dislocation with retinal involvement.
**Action [Diag]**: Lens Dislocation ✓

GPT-4.1-mini conducted a relatively long sequence of dialogue turns. Through extensive questioning, it was able to identify a considerable number of **positive findings** (8) and **negative findings** (16). This comprehensive but lengthy information-gathering process ultimately allowed the model to converge on the correct diagnosis of *lens dislocation*.

---

**Dialogue History of Gemini-2.5-Pro**

**Name**: Neurofibromatosis
**Final Diagnosis**: Neurofibromatosis Type I (NF1) ✓
**Number of Turns**: 7
**Positive Findings**: 5
**Negative Findings**: 1

**Chief Complaint Exploration (CCE)**: 4
**istory Completeness (HC)**: 2
**Evidence Chain Integrity (ECI)**: 15
**Test Justification (TJ)**: 10
**Differential Diagnosis (DDx)**: 8
**Diagnostic Correctness (DC)**: 25
**Diagnostic Uncertainty (DU)**: 2
**Diagnostic Quality Score (DQS)**: 66

CLINICAL DIALOGUE HISTORY

**Patient**: Doctor, I noticed some café-au-lait spots on my skin, and my eyelid seems a bit swollen.

**Doctor (Reasoning)**: The patient presents with café-au-lait spots and eyelid swelling. Differential considerations include: 1. **Neurofibromatosis Type I (NF1)**: Café-au-lait spots are characteristic, and eyelid swelling may correspond to a plexiform neurofibroma.
2. **Systemic diseases**: Such as nephrotic syndrome or hypothyroidism, which can cause bilateral eyelid edema but do not explain café-au-lait spots.
3. **Dermatologic conditions**: Certain skin disorders may have pigmentation and localized edema, but café-au-lait spots are highly specific.
4. **Connective tissue disorders**: E.g., dermatomyositis may cause eyelid edema, but skin rash color differs from café-au-lait spots.
To differentiate local lesions from systemic causes, eyelid swelling laterality is important. Unilateral swelling favors a localized lesion.
**Action [Ask]**: Is the eyelid swelling unilateral or bilateral?
**Patient**: My eyelid swelling is unilateral.

**Doctor (Reasoning)**: Unilateral eyelid swelling strongly suggests a local lesion, making systemic causes unlikely. The combination of café-au-lait spots and unilateral eyelid swelling highly suggests NF1. Next, it is important to assess the number and size of café-au-lait spots according to NF1 diagnostic criteria.
**Action [Ask]**: How many café-au-lait spots do you have?
**Patient**: I have about a dozen spots.

**Doctor (Reasoning)**: The patient has more than six café-au-lait spots and unilateral eyelid swelling, consistent with NF1 diagnostic criteria. To confirm, we need to check the size of the largest spot.
**Action [Ask]**: What is the diameter of the largest café-au-lait spot?
**Patient**: The largest spot is about 5 cm in diameter.

**Doctor (Reasoning)**: With over six café-au-lait spots and the largest ¿15 mm, this criterion is satisfied. Unilateral eyelid swelling likely represents a plexiform neurofibroma. Clinical examination is needed to confirm its characteristics.
**Action [Test]**: Examine the swollen eyelid.
**Examiner**: No redness, rash, or other surface abnormalities observed.

**Doctor (Reasoning)**: Surface exam excludes inflammation but does not clarify the swelling. Plexiform neurofibromas are often soft, non-compressible nodules ("bag of worms" feel). Palpation is required.
**Action [Test]**: Palpate the swollen eyelid.
**Examiner**: Swelling present; soft nodules palpable, non-tender.

**Doctor (Reasoning)**: Findings strongly suggest NF1. Café-au-lait spots ¿6 and unilateral plexiform neurofibroma meet two NF1 diagnostic criteria. For additional support, check for axillary or inguinal freckling (Crowe sign).
**Action [Test]**: Examine axillary skin.
**Examiner**: Multiple light brown macules with clear borders observed.

**Doctor (Reasoning)**: Patient now clearly meets three NF1 criteria:
1. Six or more café-au-lait spots (>15 mm).
2. Plexiform neurofibroma (unilateral eyelid swelling).
3. Axillary freckling (Crowe sign).
Since only two criteria are required for diagnosis, NF1 diagnosis is definitive.
**Action [Diag]**: Neurofibromatosis Type I (NF1). ✓

Gemini-2.5-Pro systematically collected key findings and correctly diagnosed Neurofibromatosis Type I (NF1). Our framework counts subtype-level diagnoses like this as correct.

---

### Dialogue History of Llama-4-Scout

**Answer**: rickets
**Final Diagnosis**: Autonomic nervous system dysfunction ✗

**Number of Turns**: 8
**Positive Findings**: 1
**Negative Findings**: 12

**Chief Complaint Exploration (CCE)**: 2
**istory Completeness (HC)**: 4
**Evidence Chain Integrity (ECI)**: 5
**Test Justification (TJ)**: 4
**Differential Diagnosis (DDx)**: 8

**Diagnostic Correctness (DC)**: 10
**Diagnostic Uncertainty (DU)**: 2
**Diagnostic Quality Score (DQS)**: 35

**Patient**: Recently, I noticed that my child sweats excessively on the head during sleep at night and seems constantly irritable.

**Doctor (Reasoning)**: The child's symptoms of excessive nighttime head sweating and persistent irritability could be associated with multiple conditions, such as calcium deficiency, hyperthyroidism, or tuberculosis infection. To clarify the diagnostic direction, more information on the child's basic condition and symptom details is required.
**Action [Ask]**: Has your child experienced fever, cough, or weight loss recently?
**Patient**: No fever. No cough. No weight loss.

**Doctor (Reasoning)**: The absence of fever, cough, and weight loss reduces the likelihood of tuberculosis. Given the symptoms of nighttime head sweating and irritability, calcium deficiency or hyperthyroidism is more probable. Further inquiry into the child's diet and growth is needed.
**Action [Ask]**: How has your child's diet been recently? Are they consuming sufficient dairy products or calcium supplements?
**Patient**: The child consumes little dairy and has not received additional calcium supplements.

**Doctor (Reasoning)**: Low dietary calcium intake and no supplementation increase the likelihood of calcium deficiency. Symptoms of nighttime sweating and irritability also support this diagnosis. Further testing is required to confirm.
**Action [Test]**: Serum calcium level measurement.
**Examiner**: Within normal range.

**Doctor (Reasoning)**: Normal calcium levels reduce the probability of calcium deficiency. Considering the symptoms, hyperthyroidism is now more likely. Further inquiry into other hyperthyroid symptoms is necessary.
**Action [Ask]**: Has your child experienced palpitations, hand tremors, or difficulty concentrating recently?
**Patient**: No palpitations. No hand tremors. No attention deficit.

**Doctor (Reasoning)**: The absence of typical hyperthyroid symptoms decreases the likelihood of hyperthyroidism. Nighttime sweating may be caused by multiple other conditions. Further testing is needed.
**Action [Test]**: Complete blood count (CBC) and erythrocyte sedimentation rate (ESR).
**Examiner**: Both within normal range.

**Doctor (Reasoning)**: There are no signs of infection or hyperthyroidism. Normal calcium levels and normal CBC/ESR suggest that other neurological or endocrine conditions may be involved. Further evaluation is required.
**Action [Test]**: Brain MRI.
**Examiner**: Normal.

**Doctor (Reasoning)**: Normal MRI excludes structural brain lesions. Symptoms could relate to other endocrine conditions, such as adrenal hyperactivity.
**Action [Test]**: Urinary catecholamine test.
**Examiner**: Within normal range.

**Doctor (Reasoning)**: Common endocrine and neurological causes have been largely excluded. Consideration should be given to autonomic nervous system dysfunction.
**Action [Diag]**: Autonomic nervous system dysfunction. ✕

In this consultation, Llama-4-Scout diagnosed too early, without enough positive evidence. It reached a conclusion before gathering key clinical findings, showing a tendency to rush to a diagnosis rather than collect information systematically.

---

**Dialogue History of Qwen3-8B**

**Answer**: Pneumoconiosis
**Final Diagnosis**: Pulmonary Tuberculosis ✗

**Number of Turns**: 6
**Positive Findings**: 4
**Negative Findings**: 7

**Chief Complaint Exploration (CCE)**: 4
**istory Completeness (HC)**: 6
**Evidence Chain Integrity (ECI)**: 10
**Test Justification (TJ)**: 6
**Differential Diagnosis (DDx)**: 0
**Diagnostic Correctness (DC)**: 4
**Diagnostic Uncertainty (DU)**: 1
**Diagnostic Quality Score (DQS)**: 31

**Patient**: Doctor, I have recently been experiencing shortness of breath, especially during physical activity.

**Doctor (Reasoning)**: Dyspnea worsened by exertion suggests cardiopulmonary conditions such as heart failure, coronary artery disease, or chronic obstructive pulmonary disease (COPD). It is necessary to clarify the presence of chest pain, cough, or sputum production to help differentiate the cause.
**Action [Ask]**: Do you have chest pain, cough, or sputum production?
**Patient**: Sometimes I have chest pain, especially with deep breaths or coughing. I often cough. Occasionally I produce sputum, white or gray in color.

**Doctor (Reasoning)**: Dyspnea with chest pain (related to breathing/coughing), cough, and white/gray sputum requires differentiation among pleuritis, pneumonia, COPD, or heart failure. Further clarification of infection signs (fever, night sweats) or cardiovascular risk factors (hypertension, history of heart disease) is needed to narrow the differential diagnosis.
**Action [Ask]**: Have you experienced fever, night sweats, or significant recent weight loss?
**Patient**: No fever. No night sweats. No significant recent weight loss.

**Doctor (Reasoning)**: Absence of fever, night sweats, and weight loss reduces the likelihood of infectious disease. Dyspnea with chest pain (provoked by breathing/cough) and white sputum requires focus on pleuritis, COPD, or heart failure. Need to clarify cardiovascular risk factors or history of chronic lung disease.
**Action [Ask]**: Do you have a history of hypertension, coronary artery disease, or long-term smoking?
**Patient**: No hypertension. No history of coronary artery disease. No long-term smoking history.

**Doctor (Reasoning)**: No cardiovascular risk factors. Dyspnea with chest pain still requires exclusion of myocardial ischemia or pericarditis. Chest pain related to respiration/coughing favors pleuritis or pulmonary disease. Cardiac and pulmonary evaluations are required. Prioritize ECG to exclude acute cardiac events, followed by chest imaging.
**Action [Test]**: Electrocardiogram (ECG).
**Examiner**: Sinus rhythm, normal heart rate.

---

> **Doctor (Reasoning)**: ECG excludes acute cardiac events. Pulmonary evaluation is needed. Dyspnea with chest pain and white sputum suggests pleuritis, pneumonia, or interstitial lung disease. Chest imaging should be performed to assess for inflammation, effusion, or structural abnormalities.
> **Action [Test]**: Chest X-ray.
> **Examiner**: Multiple small round nodular shadows in the upper lungs, some coalescing into large fibrotic areas.
>
> **Doctor (Reasoning)**: Chest X-ray shows multiple upper lung nodules and fibrosis. Combined with dyspnea, chest pain, and chronic cough, high suspicion for pulmonary tuberculosis or chronic infectious lung disease exists. Further evaluation for TB evidence (sputum testing, TB antibody testing) or occupational lung disease history is required. If sputum testing is not immediately available, chest CT may be prioritized to assess nodule characteristics and dissemination.
> **Action [Diag]**: Pulmonary tuberculosis. ×

Qwen3-8B incorrectly diagnosed the patient with *pulmonary tuberculosis*. The available information—multiple upper-lung nodules and fibrosis on X-ray without systemic infection signs—was insufficient for a definitive differential diagnosis. The model did not collect adequate data on occupational exposure or other relevant history, leading to a misinterpretation and misdiagnosis.

