# OpenReview forum: "Can Large Language Models Think Like Doctors? An Interactive Approach to Evaluating Clinical Reasoning"
_ICLR.cc/2026/Conference — ICLR 2026 Conference Withdrawn Submission_

### Official Review · Reviewer_yHUw · 2025-10-19

**Soundness:** 3
**Presentation:** 3
**Contribution:** 2
**Rating:** 4
**Confidence:** 4

**Summary:**

This paper proposes a benchmark for clinical reasoning in large language models (LLMs) named iClinReason, an interactive framework that simulates multi-turn doctor-patient diagnostic dialogues. Grounded in a disease knowledge graph, the framework dynamically generates structured patient cases and instantiates three agents: a Doctor Agent and an Examiner Agent. The evaluation protocol goes beyond diagnostic accuracy to include fine-grained efficiency metrics and a rubric-based diagnostic quality assessment across seven dimensions.

**Strengths:**

1. The paper identifies that static benchmarks fail to capture clinical reasoning’s interactive, iterative nature – a critical gap for deploying LLMs in healthcare.
2. The integration of accuracy, efficiency, and quality metrics provides a more holistic view of LLM performance than single-dimension benchmarks.
3. The knowledge graph-driven pipeline mitigates benchmark contamination – a major issue in LLM evaluation – by generating dynamic, unseen cases.
4. Evaluating 15 LLMs offers valuable empirical insights into performance differences.

**Weaknesses:**

1. Methodological Flaws.

1.1. Using GPT-5 for Patient/Examiner Agents creates a “black box” dependency. A single LLM’s idiosyncrasies (e.g., overspecific test results) could artificially inflate or deflate Doctor Agent scores.

1.2. The paper uses a high-performing LLM to score diagnostic quality but does not: (a) validate scores against human clinicians (e.g., calculate Cohen’s kappa); (b) test for judge bias (e.g., favoring proprietary LLMs); (c) provide the full prompt used for scoring, undermining reproducibility.

1.3. The maximum dialogue turn limit (15) and inference parameters (temperature=0.1, top-p=0.9) are selected without justification. For example, 15 turns may be insufficient for complex cases (e.g., rare diseases) or excessive for simple ones (e.g., acute sinusitis).

1.4. Synthetic cases lack clinical nuance such as: (a) comorbidities (e.g., a patient with diabetes and pneumonia); (b) vague symptom descriptions (e.g., “my chest hurts on and off” vs. structured symptom lists); (c) patient non-compliance (e.g., refusing tests). The paper acknowledges this limitation but offers no roadmap to address it.

2. Experimental Design Shortcomings.

2.1 The paper does not test the impact of individual framework components, which is the main technique flaws of the current version.

2.2 Experiments rely solely on synthetic cases. The paper provides no evidence that iClinReason’s results correlate with LLM performance on real clinical data.

2.3 Key details about evaluated models are missing, including: (a) parameter counts for proprietary models (e.g., GPT-4.1-mini); (b) fine-tuning status (e.g., whether models were pre-trained on medical data); (c) inference latency, a critical metric for clinical deployment.

2.4 The paper reports means and standard errors but no p-values for model comparisons. For example, it claims Gemini-2.5-Pro is the top-performing model but does not confirm that its accuracy (72.87%) is significantly higher than GPT-4.1-mini (71.73%).

3. Other concern.

The interactive evaluation paradigm builds on existing work in non-medical domains (e.g., MultiWOZ for task-oriented dialogue) and medical multi-agent simulations (e.g., AIPatient). The core innovation – combining knowledge graphs with multi-turn dialogue – is a incremental extension rather than a paradigm shift.

**Questions:**

In addition to the above concerns, I also have the following questions that the author needs to respond:

1. Have you validated the Patient/Examiner Agents’ outputs with board-certified physicians? Please provide inter-rater reliability scores (e.g., Cohen’s kappa) between GPT-5’s responses and human clinician judgments. Would using different LLMs (e.g., Claude-4, Gemini-2.5-Pro) for these agents change evaluation results?

2. What steps did you take to validate the “LLM-as-a-Judge” scoring? Please provide data on inter-LLM judge consistency and correlation with human clinician scores. How were the weights for the seven quality dimensions determined?

3. Why did you not compare iClinReason to established medical LLM benchmarks (e.g., MedQA, MedCaseReasoning)? Can you demonstrate that iClinReason’s metrics better predict real-world clinical performance (e.g., correlation with physician ratings of LLM outputs)?

4. What is the clinical basis for setting the maximum dialogue turn limit to 15? Have you tested how varying this threshold (e.g., 10 vs. 20 turns) impacts model performance? Similarly, why were temperature=0.1 and top-p=0.9 chosen for inference – do other parameter combinations yield different results?

---

### Official Review · Reviewer_JSvx · 2025-10-27

**Soundness:** 1
**Presentation:** 3
**Contribution:** 1
**Rating:** 0
**Confidence:** 4

**Summary:**

The authors of this paper seek to further explore the iterative process of information gathering core to clinical reasoning and diagnosis. They offer a framework for simulation of multi-turn diagnostic dialogues, and of particular novelty they seek to engage in high-quality assessment of the underlying process of clinical reasoning. They engage in a combination of simply automated metrics (e.g. number of positive findings identified) as well as more complex LLM-as-judge. They report finding errors and conclude about limitations in the clinical reasoning of current systems.

**Strengths:**

- The authors do a good job of summarizing existing issues in the reasoning assessment of large language models, and the introduction is clear and comprehensive. Some more recent work would benefit from citation, but this is contemporaneous and should not be expected (e.g.  https://ai.nejm.org/doi/full/10.1056/AIdbp2500120). They offer a reasonable summary of the relevant literature, and they are clear about their attempted contribution.
- 	- The authors are aiming at a very important paradigm with respect to automated analysis of clinical reasoning quality, and many aspects of their agentic scaffold are reasonable.
The authors test an appropriately wide range of models

**Weaknesses:**

Re: Soundness:
- As further discussed below in weaknesses, I have several major concerns regarding the soundness of this paper and its contents:
- First and most importantly, I have very little confidence in their implementation of LLM-as-judge absent substantial human validation which would be an impressive paper in its own right. While I can be convinced of LLM-as-judge in a more rote way for e.g. basic semantic matching of multiple formulations of a diagnosis, even this requires extensive human evaluation at this stage in order for it to be relied upon.
- When it comes to questions as complex as those that the authors' are asking of the model (e.g. " Key diagnostic hypothesis lacks supporting evidence" or "Final diagnosis fully consistent with all evidence and guideline-aligned"), there are numerous possible points of failure. Indeed, an auto-grader capable of performing this sort of evaluation would be a remarkable advance in the field, and in many ways would itself represent a frontier agent. Without extensive, extensive manual validation and involvement of physician scrutiny of each of these aspects, I fear that this metric is primarily noise. I have associated concerns about the rubric itself, as the calibration and exclusivity of each of these categories is not clear.
- In addition to my concerns about the LLM-as-judge version used, which I believe are themselves disqualifying, I have further concerns about the multi-agent setups common to this line of research. Specifically, the generation of "plausible" information about tests etc will always carry the risk of either information leakage, or the generation of inappropriate info.
- Finally, I am concerned that the authors' heavy simplifications of the nature of clinical reasoning elide important criteria (e.g. resource stewardship, and the recognition of pertinent negatives).
- Ultimately, neither end of the authors' approach to evaluating clinical reasoning satisfies me. Their simple automated methods are too abstracted, and their LLM-as-judge is incompletely validated. Lacking this, the novelty and quality of the paper is limited.

RE: Contribution:
Overall, this paper remains very incremental with respect to the field, offering a repeated instantiation of the well-trod "LLM-as-judge" paradigm. If the authors did achieve their goals with respect to automated evaluation of complex clinical reasoning it would be incredibly novel, however due to my aforementioned concerns I don't think they do this.

WEAKNESSES (detailed):
- I have a fundamental concern that persists with this paper, as it has been present in most of the "multi-agent simulation" literature, which is that the use of a "patient LLM" has substantial risks of leaking information to the "clinician LLM", or generating inappropriate-yet-plausible information when the clinician LLM goes beyond the boundaries of the expected interaction. This is not an absolute limitation, and this paper is a reasonable contribution within this aspect of the literature despite it, but it does need to be clearly acknowledged.
- There also appears to be a clear tension between the authors' assertion that (1) their system is capable of dynamically generating an arbitrary amount of cases, and (2) these cases remain firmly grounded in clinical knowledge. At the very least, there is a clear tension between the associated breadth of cases and how well-constrained they may be. Using LLMs to solely serve information that has been pre-defined into a conversational format is one thing, but whenever generation occurs there are several risks including generation of physiologically incorrect info, and leakage of the final diagnosis via specific language and stereotyping of responses.
- The authors make several statements downplaying this possibility, e.g. "
test results generated by AE reflect medically plausible outcomes tied to d. This design guarantees that any observed errors in
reasoning are attributable to the Doctor Agent rather than noise in the environment." however, this cannot be the case as their statement clearly makes clear. "Medically plausible" remains a source of noise.
- While I appreciate the authors' attempts at abstraction of the clinical reasoning process, several of their abstractions and evaluation methods are not aligned with clinical reality. For example, "total dialogue turns" are only relevant in cases where the model has a correct final diagnosis. Being efficiently wrong is not helpful to patients. Similarly, while they emphasize "positive findings per turn", pertinent negative findings are just as important and often more important, a good, harm-avoiding clinician should be aware of this as should a good agent.
- With respect to diagnostic quality, LLM-As-Judge cannot be relied upon in this context, unless clearly validated against clinician judgment. The LLM-As-Judge is not sufficiently described or validated in this paper to be effectively relied upon, especially for these more complex tasks and aspects of diagnostic reasoning.

**Questions:**

- How was this diagnostic quality score generated? Was any substantial validation of this done against human assessments? What is the relevant calibration? The score is from 1-100, has any validation been done to show that this is sufficiently granular?
- Has any evaluation of the severity of errors been performed in detail?

---

### Official Review · Reviewer_zDNx · 2025-11-01

**Soundness:** 3
**Presentation:** 3
**Contribution:** 2
**Rating:** 6
**Confidence:** 3

**Summary:**

The paper presents a method for dynamically generating patient-doctor interactions for evaluating a model’s ability to identify the disease that a patient has as a doctor. The method first generates (or samples) a patient profile from knowledge graphs and textual knowledge using an LLM, where the profile contains a specific disease to be diagnosed, relevant descriptive passages, sampled patient attributes, and symptom manifestations. From the profile, the patent agent starts a conversation with a doctor agent to be evaluated. The doctor agent can request a medical test, which is then generated from the profile by an examiner agent. The paper also proposes three evaluation metrics. The paper also presents evaluation results of some recent LLMs using the proposed evaluation framework.

**Strengths:**

1. I think this is an interesting paper, providing a simple, practical, and realistic way to evaluate LLMs in terms of medical reasoning capability. I believe this paper contributes to the community.
2. The paper is clearly written and easy to follow.

**Weaknesses:**

1. The major problem of the paper is the validity of the patient and examiner agents’ output. The paper provides some experimental results on how current LLMs serve as a doctor agent, but it’s not clear how patient and examiner agents are realistic. Quantitative evaluations with some objective criteria are desirable. At least, some subjective evaluations by human experts to see the validity of the patient and examiner agents are necessary.
2. Related to 1, I cannot see how the paper incorporates the ambiguity of real patients and the uncertainty or randomness of examinations.
3. A more specific task definition is preferable. As the profile is generated from the knowledge that is available for doctors, it also makes sense to allow a model to use the same knowledge as that of the model for case generation uses. In this case, an LLM’s abilities to select appropriate actions and generate appropriate text are evaluated. Meanwhile, the experimental results seem to only show the case when the knowledge is not available and LLMs’ internal knowledge is used. The paper’s position may be clearer if it explicitly mentions what aspects the method tries to evaluate and if it gives some rationale for the choice.
4. Related to 3, it would be nice if the paper could provide the performance when the LLMs have access to the knowledge. This can provide some insights into the challenges in this evaluation method (i.e., whether it lies in knowledge or in the reasoning).

**Questions:**

I would like to see some responses from the authors regarding the weaknesses.

---

### Official Review · Reviewer_YHKm · 2025-11-04

**Soundness:** 2
**Presentation:** 3
**Contribution:** 2
**Rating:** 2
**Confidence:** 5

**Summary:**

The key contributions of this paper according to the authors are an interactive evaluation framework for clinical reasoning, knowledge-grounded case generation and process-aware assessment, and comprehensive analysis of LLMs’ diagnostic behaviors. They tested 15 AI models (both proprietary and open source) on their diagnostic reasoning task and found that most models fail in clinical reasoning.

**Strengths:**

-The authors rightly noted that models had trouble with information inquiry. For example, for the one case shown,  no questions regarding the patient's medical history (such as smoking or COPD) that would be relevant were asked.
-They tested a significant number of models

**Weaknesses:**

-The paper's major contribution of using an interactive framework for assessing clinical reasoning was previously thoroughly explored in other work ("An evaluation framework for clinical use of large language models in patient interaction tasks") that is not mentioned: https://www.nature.com/articles/s41591-024-03328-5. This diminishes one of the major claimed contributions of the work.
-"Total Dialogue Turns" were a metric that was assessed, but looking at the single case shown -- the signs and symptoms were not at all atomic. The patient came in presenting significant more information than a real patient might. The patient basically gave a one liner history for community pneumonia that didn't require much further inquiry.
-I am concerned that the knowledge graph cases, while shown to be "accurate" by physician assessment do not actually present atomic signs and symptoms which would require further inquiry from the physician agent. The patient is likely revealing too much too quickly.
-No overlap in physician graded cases to assess inter-grader agreement.
-No information on exactly how physicians assessed cases produced by knowledge graph
-Regarding information leakage, while the diagnosis may not have been shared, the first turn may have given away too much information leading to diagnosis too quickly and in a way that is unrealistic. This would also impact how model efficiency is assessed.
-The knowledge graph is built off of publicly available data that could be in the training data of the models, impacting whether or not their performance on the test set is actually independent of the training set.

**Questions:**

What were the backgrounds and experiences of the grading physicians? How were they asked to grade?

---

### Note · Authors · 2025-11-23

I have read and agree with the venue's withdrawal policy on behalf of myself and my co-authors.